# Self-supervised Learning for Segmentation and Quantification of Dopamine Neurons in Parkinson's Disease

**Fatemeh Haghighi**                                                 *fhaghigh@asu.edu*

*Arizona State University, Tempe, AZ 85281, USA*
*Department of Neuroscience, Genentech Inc., South San Francisco, CA 94080, USA*

**Soumitra Ghosh**[*]                                                *ghoshs29@gene.com*
*Department of Neuroscience, Genentech Inc., South San Francisco, CA 94080, USA*

**Hai Ngu**                                                          *hain@gene.com*
*Department of Pathology, Genentech Inc., South San Francisco, CA 94080, USA*

**Sarah Chu**                                                        *chus18@gene.com*
*Department of Neuroscience, Genentech Inc., South San Francisco, CA 94080, USA*

**Han Lin**                                                          *hanhlin@gene.com*
*Department of Neuroscience, Genentech Inc., South San Francisco, CA 94080, USA*

**Mohsen Hejrati**                                                   *hejratis@gene.com*
*Department of Artificial Intelligence, Genentech Inc., South San Francisco, CA 94080, USA*

**Baris Bingol**                                                     *barisb@gene.com*
*Department of Neuroscience, Genentech Inc., South San Francisco, CA 94080, USA*

**Somaye Hashemifar**[*]                                             *hashems4@gene.com*
*Department of Artificial Intelligence, Genentech Inc., South San Francisco, CA 94080, USA*

**Reviewed on OpenReview:** *https://openreview.net/forum?id=izFnURFG3f*

## Abstract

Parkinson's Disease (PD) is the second most common neurodegenerative disease in humans. PD is characterized by the gradual loss of dopaminergic neurons in the Substantia Nigra (SN, a part of the mid-brain). Counting the number of dopaminergic neurons in the SN is one of the most important indexes in evaluating drug efficacy in PD animal models. Currently, analyzing and quantifying dopaminergic neurons is conducted manually by experts through analysis of digital pathology images which is laborious, time-consuming, and highly subjective. As such, a reliable and unbiased automated system is demanded for the quantification of dopaminergic neurons in digital pathology images. Recent years have seen a surge in adopting deep learning solutions in medical image processing. However, developing high-performing deep learning models hinges on the availability of large-scale, high-quality annotated data, which can be expensive to acquire, especially in applications like digital pathology image analysis. To this end, we propose an end-to-end deep learning framework based on self-supervised learning for the segmentation and quantification of dopaminergic neurons in PD animal models. To the best of our knowledge, this is the first deep learning model that detects the cell body of dopaminergic neurons, counts the number of dopaminergic neurons, and provides characteristics of individual dopaminergic neurons as a numerical output. Extensive experiments demonstrate the effectiveness of our model in quantifying neurons with high precision, which can provide a faster turnaround for drug efficacy studies,

---

[*]Corresponding authors

better understanding of dopaminergic neuronal health status, and unbiased results in PD pre-clinical research. As part of our contributions, we also provide the *first* publicly available dataset of histology digital images along with expert annotations for the segmentation of TH-positive DA neuronal soma.

# 1 Introduction

Parkinson's Disease (PD) is the second most common neurodegenerative disease in humans (Poewe et al., 2017). PD is characterized by the gradual loss of dopaminergic neurons (DA neurons) in the Substantia Nigra (SN). SN is the area of the midbrain that consists of DA neurons which are most susceptible to genetic and sporadic factors and lost in PD. Loss of dopaminergic neurons leads to motor neuron associated dysfunctions as observed in PD patients and animal models (Johnson et al., 2012). Preventing loss of DA neurons is the most important goal of PD targeting therapies. Immunostaining of Tyrosine Hydroxylase (TH), an enzyme expressed predominantly in DA neurons, is the most reliable method used for detecting DA neurons, and the TH staining intensity is also an indicator of the health status of the DA neurons (Ghosh et al., 2021). Given these factors, pre-clinical research on PD is highly dependent on the segmentation and quantification of DA neurons in the SN (Poewe et al., 2017; Guatteo et al., 2022). Currently, analyzing and quantifying dopaminergic neurons is conducted manually by experts through analysis of digital pathology images, which is laborious, time-consuming, and highly subjective. As such, reliable and unbiased automated systems are highly demanded for the quantification of dopaminergic neurons in digital pathology images. Such systems will make a significant impact in the field of PD pre-clinical research by identifying the efficacy of potent drugs in a shorter time-frame and accelerating the possibility of taking a potential drug into the clinic.

Recent years have seen a surge in adopting deep learning solutions in various medical image processing applications, including cell segmentation in microscopy images. Numerous studies have developed deep learning-based cell segmentation methods that are specialized for images with large-scale training datasets (Falk et al., 2019). However, the scarcity and difficulty in accessing labeled data for DA neurons pose a significant challenge to the success of these methods for segmentation and quantification of DA neurons. Moreover, due to the unique morphology and organization of DA neurons, the efficiency of generalist cell segmentation methods (Stringer et al., 2021) for segmentation of DA neurons may be limited (Robitaille et al., 2022). Furthermore, generalist cell segmentation methods fall short in providing specific information about the characteristic of DA neurons, which holds substantial value in PD research. Therefore, it is crucial to develop specialized machine learning models that can analyze and quantify DA neurons precisely in the SN.

To address these challenges, we propose an end-to-end deep learning framework for the segmentation and quantification of dopaminergic neurons in the SN of mouse brain tissues. To the best of our knowledge, this is the first deep learning model that segments the cell body of dopaminergic neurons, counts the number of dopaminergic neurons, and provides characteristics of individual dopaminergic neurons as a numerical output. Particularly, to overcome the labeled data scarcity challenge, our proposed method leverages cross-domain self supervised learning (Zbontar et al., 2021; Azizi et al., 2021; Caron et al., 2021; Haghighi et al., 2021; 2022) on both large-scale unlabeled natural images and domain specific pathology images. The self-supervised pre-trained models are further fine-tuned for the DA neuron segmentation using limited labeled data. Moreover, we propose a practical approach for counting the number of DA neurons from the segmented cells. Our extensive experiments demonstrate that our cross-domain self-supervised learning provides superior segmentation accuracy compared to (1) training segmentation models from scratch (random initialization), (2) fine-tuning conventional supervised ImageNet models, and (3) fine-tuning self-supervised pretrained models on just natural or in-domain datasets. Additionally, our experiments highlight the efficacy of our method in counting the number of DA neurons, which can provide a faster turnaround for drug efficacy studies, better understanding of dopaminergic neuronal health status, and unbiased results in PD pre-clinical research.

We provide the first publicly available dataset of digital microscopy images along with expert annotations for the segmentation of TH-positive DA neurons in mouse brains. We anticipate that this novel dataset will expedite the development and assessment of new machine learning models for the segmentation of DA neurons in digital pathology images, further advancing research in Parkinson's disease.

In summary, we make the following contributions:

- The first publicly available dataset of digital microscopy images along with expert annotations for the segmentation of TH-positive DA neurons.

- The first end-to-end framework for automatic segmentation and quantification of DA neurons in whole-slide digital pathology images of PD models.

- A cross-domain self-supervised pre-training approach that exploits the power of unlabeled natural and medical images for representation learning.

- A comprehensive set of experiments that demonstrate the effectiveness of our method in segmenting and quantifying DA neurons using a limited amount of annotated data, and providing additional information about phenotypic characteristics of DA neurons.

## 2 Related Works

**CNN-based quantification of dopaminergic neurons.** The current strategies for analyzing DA neurons face significant challenges. Traditional stereology is extremely time consuming and subjective to user associated bias. Moreover, threshold for optical density based method cannot capture the complex nature of DA neurons on histology sections and provide inaccurate numbers. Recently, deep learning methods have been successfully utilized in analyzing human digital pathology images for different tasks, including cell segmentation and cell counting Ronneberger et al. (2015); Falk et al. (2019); Greenwald et al. (2022); Moshkov et al. (2020); Hatipoglu & Bilgin (2017). Nucleus segmentation methods (Hollandi et al., 2022) detect nuclei of cells, but they fail to detect individual cells and cannot retrieve the information on cell body (i.e. TH intensity in soma) which can be used to interpret the biology of DA neurons. Generalist cell segmentation models such as Cellpose (Stringer et al., 2021) have been developed to segment many types of cell. Cellpose relies on a large dataset of images of different cells and a reversible transformation from the training set segmentation masks to vector gradients that can be predicted by a neural network. In particular, it leverages a U-Net model to predict the horizontal and vertical gradients, as well as whether a pixel belongs to any cell. The three predicted maps are then combined into a gradient vector field. The predicted gradient vector fields are used to construct a dynamical system with fixed points whose basins of attraction represent the predicted masks. Despite its success, the efficiency of Cellpose in detecting specific types of neurons such as DA neurons is still limited (Robitaille et al., 2022). The number of studies that employ deep learning for the quantization of DA neurons in animal models of PD is relatively limited. Penttinen et al (Penttinen et al., 2018) implemented a deep learning-based method for processing whole-slide digital imaging to count DA neurons in SN of rat and mouse models. This study leverages the TH positive nucleus to detect the TH cells which is susceptible to error because of the architecture of DA neurons in SN and makes it difficult to distinguish between overlapping cells when detected only relying on the nucleus as annotations. Zhao et al (Zhao et al., 2018) developed a framework for the automatic localization of SN region and detection of neurons within this region. The SN localization is achieved by using a Faster-RCNN network, whereas neuron detection is done using a LSTM network. However, these studies are limited to counting neurons and/or detecting neuron locations and do not provide additional information about individual cells, such as TH intensity and health status, which is essential for understanding the biology behind DA neuronal loss and its association with PD pathogenesis.

**Self-supervised learning.** Self-supervised learning methods aim to learn generalizable representations directly from unlabeled data. This paradigm involves training a neural network on a manually created (pretext) task for which ground truth is obtained from the data. The learned representations can be transferred and fine-tuned for various target tasks with limited labeled data. Instance discrimination methods (Zbontar et al., 2021; He et al., 2020; Caron et al., 2021; Grill et al., 2020; Chen & He, 2021; Chen et al., 2020) have recently sparked a renaissance in the SSL paradigm. These methods consider each image as a separate class and seek to learn representations that are invariant to image distortions. Contrastive approaches, such as MoCo (He et al., 2020), consider the augmented views of the same image as positive pairs and the ones from the other images as negative pairs and aim to enhance the similarity between positive pairs while decreasing

the similarity between negative pairs. Barlow Twins (Zbontar et al., 2021) minimizes redundancy in the representations by measuring the cross-correlation matrix between the embeddings of two views of the same image and making it close to the identity matrix. Clustering approaches, such as deep cluster (Caron et al., 2021) and SwAV (Caron et al., 2021), simultaneously cluster the data while enforcing agreement between cluster assignments generated for different views of the same image. Motivated by their success in computer vision, instance discrimination SSL methods have been adopted in medical applications. A systematic transfer learning study for medical imaging (Hosseinzadeh Taher et al., 2021) demonstrated the efficacy of existing instance discrimination methods pre-trained on ImageNet for various medical tasks. A group of works focused on designing SSL frameworks by exploiting consistent anatomical structure within radiology scans (Taher et al., 2023; Haghighi et al., 2021; 2020). Another line of studies designed contrastive-based SSL for medical tasks (Hosseinzadeh Taher et al., 2022; Haghighi et al., 2022; Azizi et al., 2021; Kaku et al., 2021), including whole slide image classification (Li et al., 2021). In contrast to the previous works, our work is the first study that investigates the efficacy of SSL for digital pathology images of PD animal models to compensate for the lack of large-scale annotated datasets for training deep learning models.

## 3 Dataset

The dopaminergic neuronal soma dataset used in this study is an internal dataset of Genentech and was obtained by manually labeling approximately 18,193 TH positive DA neuronal soma in 2D histology digital images. The digital images were obtained from different animal studies where mouse brains were sectioned at 35 micron thickness and stained with TH and either Haematoxylin or Nissl as a background tissue stain. Each animal study consisted of multiple animals and all the animals in one study were stained at one timeframe. This allowed us to take into account the minor tissue processing and staining associated variability that commonly occurs in histology studies. The sections were then imaged using a whole slide scanner microscope, Nanozoomer system (Hamamatsu Corp, San Jose, CA) at 20x resolution (0.46 microns/pixel). Whole coronal brain section images containing the SN were exported from the digital scans at 20x resolution and were used to annotate the TH positive DA neuronal soma and train the model. The ground truth (GT) for this study was labeled and quality-controlled by biologists who specialize in mouse brain anatomy and PD research. We have evaluated human annotator bias in one of our previous neuroanatomy segmentation studies that have been published (Barzekar et al., 2023). We observed a 5-8 percent difference in output with different manual annotators. Since manual annotation and counting of individual cells is extremely laborious in a dataset used in this study, to mitigate the human annotator bias, we had 3 annotators who randomly annotated the cells for the GT in this study. A neuroanatomy expert further QCd the annotations and made the necessary changes to reflect the cell annotation. The representative image is depicted in Fig 2 ground truth panel. The blind test dataset (out-study test set) used for analyzing the model's efficiency was a separate animal study in which the model has not been directly trained on the study group. This dataset consists of 20 brain sections randomly selected from a different animal study with 150 brain sections. The ground truth (GT) for the out-study test set was established by annotating the soma. Our model and the baseline method (i.e. Cellpose) were tested on these 20 sections. Our cell annotation dataset consists of 114 8-bit RGB images with associated 16-bit cell label images. For the label images, each cell area is assigned a pixel value in the range of 1 to n, where n is the total number of labeled cells in a label image. Since the label images are in 16-bit, they may appear black; pre-processing (for example adjusting brightness/contrast) is needed to see the cells. For all the images, the image resolution is 0.46 microns per pixel. For facilitating future research into PD, we have publicly released the dataset, which can be found at `https://data.mendeley.com/datasets/8phmy565nk/1`.

## 4 Method

In this section, we provide the details of our deep learning framework, including the self-supervised pre-training, fine-tuning stage, and cell counting.

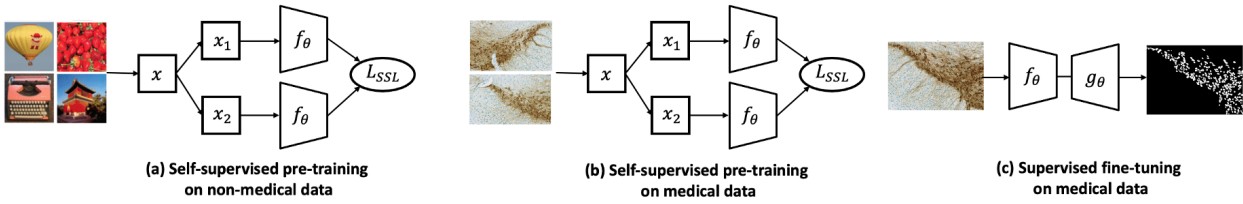

Figure 1: An overview of our approach. To address the annotated data scarcity challenge for training deep models, we perform (a) self-supervised pre-training on natural images, and then (b) self-supervised pre-training on digital pathology images. We finally (c) fine-tune the self-supervised pre-trained model with limited annotated data for the target neuron segmentation task.

## 4.1 Self-supervised pre-training

Our approach is established on continual self-supervised pre-training in which a model is first pre-trained on a massive general dataset, such as ImageNet, and then pre-trained on domain-specific datasets. For the first step (see Figure 1.a), we leverage the self-supervised models pre-trained on the ImageNet dataset using state-of-the-art instance discrimination approaches, such as Barlow Twins (Zbontar et al., 2021). For the second step (see Figure 1.b), we continue the self-supervised pre-training on the in-domain medical dataset. Finally, we fine-tune the pre-trained models for the neuron segmentation (target) task using labeled images (see Figure 1.c).

**Barlow Twins (Zbontar et al., 2021).** This SSL approach aims to reduce the amount of redundant information about each sample in the learned representations while simultaneously making the representation invariant to image distortions. To do so, given an image sample $X$, two distorted views of the sample are generated by applying a data augmentation function $\mathcal{T}(.)$ on $X$. The two distorted views $X_1$ and $X_2$ are then processed by the backbone network $f_\theta$ to produce latent representations $Z_1 = f_\theta(\mathcal{T}(X_1))$ and $Z_2 = f_\theta(\mathcal{T}(X_2))$. The backbone network $f_\theta$ includes a standard ResNet-50 encoder and a three-layer MLP projection head. The model is trained by minimizing the following loss function:

$$\mathcal{L}_{SSL} = \sum_i (1 - \mathcal{C}_{ii})^2 + \lambda \sum_i \sum_{i \neq j} \mathcal{C}_{ij}^2 \tag{1}$$

where $\mathcal{C}$ is the cross-correlation matrix computed between $Z_1$ and $Z_2$ along the batch dimension. $\lambda$ is a coefficient to identify the weight of each loss term. The model is trained by making the cross-correlation matrix $\mathcal{C}$ close to the identity matrix. In particular, by equating the diagonal elements of the $\mathcal{C}$ to 1, the learned representation will be invariant to the image distortions. By equating the off-diagonal elements of the $\mathcal{C}$ to 0, the different elements of the representation will be decorrelated, so that the output units contain non-redundant information about the images.

## 4.2 Network architecture

For self-supervised pre-training, we use a standard ResNet-50 as the backbone. For the target dopamine neuron segmentation task, we employ a U-Net network which consists of an encoder ($f_\theta$) and decoder ($g_\theta$). The encoder is a standard ResNet-50, which is initialized with the self-supervised pre-trained encoder. The decoder is initialized randomly.

## 4.3 Data sampling and augmentation

For the target dopamine neuron segmentation task, we divide images into non-overlapping patches of size $512 \times 512$ to ensure we sample from every part of the image. In all experiments, the raw image intensities per channel are normalized to the [0,1] range. Data augmentation is essential for biological and medical image analysis due to the typically limited amount of available annotated data. As such, we use different data augmentation techniques to enforce the model to capture more robust and generalizable representations.

| Pre-training | Initialization | Dice(%) |
|---|---|---|
| - | Random | 86.43±0.96 |
| Supervised | ImageNet | 85.86±3.37 |
| Self-supervised | DeepCluster-v2 | 87.13±0.69 |
| | Barlow Twins | 87.24±0.75 |
| | SwAV | **87.73±0.68** |

(a) Fine-tuning with 100% of data: the best self-supervised method (i.e SwAV) yields significant boost ($p < 0.05$) compared with the best baseline (i.e. training from random initialization).

| Pre-training | Initialization | Dice(%) |
|---|---|---|
| - | Random | 67.22±8.24 |
| Supervised | ImageNet | 76.76±4.25 |
| Self-supervised | DeepCluster-v2 | 78.72±3.98 |
| | Barlow Twins | 79.50±2.02 |
| | SwAV | **80.83±1.17** |

(b) Fine-tuning with 25% of data: the best self-supervised method (i.e SwAV) yields significant boost ($p < 0.05$) compared with the best baseline (i.e. supervised ImageNet model).

Table 1: Comparison of different initialization methods on the dopamine neuron segmentation task.

In particular, based on our ablation study, we use Flip, Rotation, RGBShift, Blur, GaussianNoise, and RandomResizedCrop to teach the expected appearance and color variation to the deep model.

### 4.4 Fine-tuning protocol

We initialize the encoder of the target model (i.e. U-Net) with the pre-trained models and fine-tune all target model parameters. We train the target models using the Adam optimizer with a learning rate of $1e-3$ and $(\beta_1, \beta_2) = (0.9, 0.999)$. We employ ReduceLROnPlateau learning rate decay scheduler. We use a batch size of 32 and train all models for 200 epochs. We employ the early-stop mechanism using the validation data to avoid over-fitting. We use the Dice coefficient loss function for training the target task. The mean dice metric is used for evaluating the accuracy of the target segmentation task. We run each method ten times on the target task and report the average and standard deviation performance over all runs as well as statistical analyses based on independent two-sample t-test.

### 4.5 Cell counting

A naive method for automatic counting of cells from segmentation predictions involves computing the connected components within the masks and considering the number of connected components as the cell count. However, this approach may not be accurate due to the presence of overlapping cells that share boundaries. To address this issue and improve counting accuracy, we first calculate the minimum and average cell size using the ground truth for the training data. Then, we take the model's predictions (segmentation masks) and extract the connected components within them; each connected component represents one or more cells (in the case of overlapping cells). We then filter out components that are smaller than the minimum cell size. For the remaining components, we count cells by dividing the cell size by the average cell size.

## 5 Experiments and Results

### 5.1 Self-supervised models provide more generalizable representations than supervised pre-trained models

**Experimental setup.** In this experiment, we evaluate the transferability of three popular SSL methods using officially released models, including DeepCluster-v2 (Caron et al., 2021), Barlow Twins (Zbontar et al., 2021), and SwAV (Caron et al., 2021). All SSL models are pre-trained on the ImageNet dataset and employ a ResNet-50 backbone. As the baseline, we consider (1) training the target model from random initialization (without pre-training) and (2) transfer learning from the standard supervised pre-trained model on ImageNet, which is the *de facto* transfer learning pipeline in medical imaging (Hosseinzadeh Taher et al., 2021). Both baselines benefit from the same ResNet-50 backbone as the SSL models.

**Results.** Table 1a displays the results, from which we draw the following conclusions: (1) transfer learning from the supervised ImageNet model lags behind training from random initialization. We attribute this inferior performance to the remarkable domain shift between the pre-training and target tasks. In

| Pre-training Method | Pre-training Dataset | Dice(%) |
|---|---|---|
| Random | - | 67.22±8.24 |
| Barlow Twins | ImageNet | 79.50±2.02 |
| SwAV | ImageNet | 80.83±1.17 |
| Barlow Twins | In-domain | 70.92±5.41 |
| Barlow Twins | ImageNet→In-domain | **81.73±1.03** |

Table 2: Comparison of pre-training dataset for self-supervised learning.

particular, supervised ImageNet models are encouraged to capture domain-specific semantic features, which may be inefficient when the pre-training and target data distributions are far apart. Our observation is in line with recent studies (Raghu et al., 2019) on different medical tasks suggests that transfer learning from supervised ImageNet models may offer limited performance gains when the target dataset scale is large enough to compensate for the lack of pre-training. (2) Transfer learning from self-supervised models provide superior performance compared with both training from random initialization and transfer learning from the supervised ImageNet model. In particular, the best self-supervised model (i.e. SwAV) yields 1.3% and 2.27% performance boosts compared with training from random initialization and the supervised ImageNet model, respectively. Our statistical analysis based on independent two-sample t-test demonstrates the significance ($p < 0.05$) of the gain provided by SwAV compared with random initialization. Intuitively, self-supervised pre-trained models, in contrast to supervised pre-trained models, encode features that are not biased to task-relevant semantics, providing improvement across domains. Our observation in accordance with previous studies (Hosseinzadeh Taher et al., 2021) demonstrates the effectiveness of self-supervised ImageNet models for medical applications.

### 5.2 Self-supervised models provide superior performance in limited data settings

**Experimental setup.** We conduct further experiments to evaluate the advantage that self-supervised pre-trained models can provide for small data regimes. To do so, we randomly select 25% of the training data and fine-tune the self-supervised pre-trained models on this subset of data. We then compare the performance of self-supervised models with training the target model from random initialization and fine-tuning the supervised ImageNet model.

**Results.** The results are shown in Table 1b. First, we observe that transfer learning from either supervised or self-supervised pre-trained models offer significant ($p < 0.05$) performance improvements compared with training from random initialization. In particular, the supervised ImageNet model provides a 9.5% performance improvement compared to the random initialization of the target model. Moreover, self-supervised models– DeepCluster-v2, Barlow Twins, and SwAV, offer 11.5%, 12.3%, and 13.6% performance boosts, respectively, in comparison with random initialization. These observations imply the effectiveness of pre-training in providing more robust target models in low data regimes. Second, we observe that self-supervised models provide significantly better performance than the supervised ImageNet model. Specifically, DeepCluster-v2, Barlow Twins, and SwAV achieve 1.96%, 2.74%, and 4% performance boosts, respectively, compared to the supervised ImageNet baseline. Our statistical analysis based on independent two-sample t-test demonstrates the significance ($p < 0.05$) of the gains provided by the best self-supervised models (i.e. SwAV and Barlow Twins) compared with the supervised ImageNet model. These observations restate the efficacy of self-supervised models in delivering more generic representations that can be used for target tasks with limited data, resulting in reduced annotation costs.

### 5.3 Impact of pre-training data on self-supervised learning

**Experimental setup.** We investigate the impact of pre-training datasets on self-supervised learning. To do so, we train Barlow Twins on three data schemas, including (1) SSL on the ImageNet dataset, (2) SSL on the medical dataset (referred to as the in-domain), and (3) SSL on both ImageNet and in-domain datasets

| Metric | Score (%) |
|---|---|
| Precision | 95.25 |
| Recall | 95.49 |
| F1-score | 95.31 |

(a) The results for counting precision, recall and F1-score of our method vs. human observers.

| Method | Counting Error (%) |
|---|---|
| Connected components | 21.66 |
| Our approach | **9.08** |

(b) The results for automatic neuron counting error compared with expert human counting.

Table 3: Dopamine neuron detection and counting results

(referred to as ImageNet→In-domain). For ImageNet→In-domain pre-training, we initialize the model with SwAV pre-trained on ImageNet, followed by SSL on our in-domain dataset. We fine-tune all pre-trained models for the neuron segmentation task using 25% of training data.

**Results.** Table 2 shows the segmentation accuracy measured by the Dice score (%) for different pretraining scenarios. First, we observe that pre-training on only in-domain dataset yields lower performance than pre-training on only the ImageNet dataset. We attribute this inferior performance to the limited number of in-domain pre-training data compared with the ImageNet dataset (1500 vs. 1.3M). Moreover, we observe that the best performance is achieved when both ImageNet and in-domain datasets are utilized for pre-training. In particular, ImageNet→In-domain pre-training surpasses both in-domain and ImageNet pre-trained models. Our statistical analysis reveals that pretraining on ImageNet followed by pretraining on In-domain data provides significant boosts ($p < 0.05$) compared with both in-domain pretraining and ImageNet pretraining standalone. These results imply that pre-training on ImageNet is complementary to pre-training on in-domain medical datasets, resulting in more powerful representations for medical applications.

## 5.4 Dopamine neurons detection and counting

**Experimental setup.** The DA neurons segmented by the model were compared to the DA neurons detected by a biologist in the same tissue section from the blind data-set. The biologist detected the DA neurons and counted them manually on an image analysis platform ImageJ. The output from the model was overlaid with the manually detected cells and based on the color coding of the DA neurons by the model, the true positive (TP), false positive (FP) and false negative (FN) were calculated by the biologist. We calculated precision, recall and F1-score metrics for the detected neurons in the test images. In these measures, TP is the number of neurons successfully detected by the model; FP is the number of neurons detected by the model but are not actually neurons; and FN is the number of neurons not detected by the model. We further compare the performance of our method in neuron counting to human counting. To do so, we calculate the percentage error between the total number of neurons counted by our method and human counting. We also conduct an ablation study to illustrate the superiority of our cell counting method over the naive approach of counting cells by the number of connected components in the images.

**Results.** The performance metrics for neuron detection are shown in Table 3a. As seen, our method can effectively detect dopaminerginc neurons in whole-slide digital pathology images; in particular, our approach achieves a precision, recall, and F1-score of 95.25%, 95.49%, and 95.31%, respectively. Moreover, Table 3b presents the neuron counting results against human counting. As seen, automatic counting of the cells through computing the connected components within segmentation masks yields an error rate of 21.66%, while incorporating the connected components' sizes in counting significantly decreases the error rate to 9%. These results demonstrate the effectiveness of our approach in handling overlapping neurons and providing a reliable automatic system for neuron counting.

## 5.5 Comparison with state-of-the-art cell counting methods

**Experimental setup.** We compare our model's efficacy in counting the dopamine neurons with the latest generalist cell segmentation model, i.e. Cellpose (Stringer et al., 2021). To do so, we first employ zero

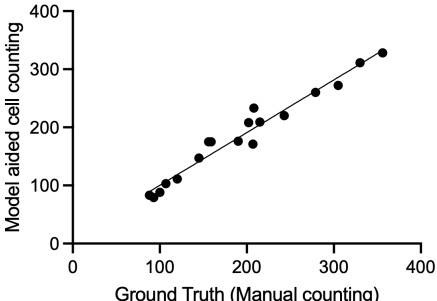

| Pearson r | |
|---|---|
| r | 0.9793 |
| 95% confidence interval | 0.9442 to 0.9924 |
| $R^2$ | 0.9591 |
| P-value | |
| P (two-tailed) | < 0.0001 |
| P-value summary | **** |
| Significant (alpha=0.05) | Yes |

Figure 2: Correlation plot depicting the number of DA neurons counted by a biologist vs. the number of DA neurons counted by our developed model. A blind data set was used to count the neurons from 18 brain sections (i.e. the number of XY pairs used in the plot) stained with TH staining to identify the DA neurons and Nissl stain to stain the brain tissue. The sections for this study were chosen from multiple animal studies.

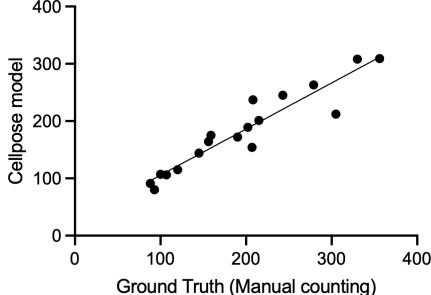

| Pearson r | |
|---|---|
| r | 0.9443 |
| 95% confidence interval | 0.8538 to 0.9794 |
| $R^2$ | 0.8917 |
| P-value | |
| P (two-tailed) | < 0.0001 |
| P-value summary | **** |
| Significant (alpha=0.05) | Yes |

Figure 3: Correlation plot depicting the number of DA neurons counted by a biologist vs. the number of DA neurons counted by the Cellpose model. A blind data set was used to count the neurons that were previously used to analyze model efficiency in Figure 2.

padding to make the size of the test images equal to a power of 512. Then, we divide the test images into non-overlapping 512×512 patches and then feed patches to the network. We then assemble the model's predictions for image patches to generate the prediction for the whole image. To examine the model's efficiency in counting DA neurons, a biologist counted the cells manually, referred to as counting ground truth (GT), in the same section (blind dataset). We then ran correlation statistics to measure the $R^2$ between the predictions of our model and the GT. We additionally compared the GT to the Cellpose's results via correlation statistics. Finally, the counting results for DA neurons from our model, GT, and the Cellpose were plotted head to head to examine the efficiency of our model.

**Results.** Figure 2 shows the correlation plot between the GT and our model's counted DA neurons. $R^2$ of 0.95 with a $p-$value $< 0.0001$ was achieved by our model in correlation statistical analysis. Figure 3 depicts the correlation plot between the GT and Cellpose. As seen, under the same parameters and dataset, Cellpose achieved a $R^2$ of 0.89 with a $p-$value $< 0.0001$ in the correlation statistical analysis. A deeper analysis of the data in Figure 4 revealed that Cellpose undercounted the DA neurons in three sections highlighted by an arrow when compared to the GT counts. By contrast, our model was able to count the cells with higher accuracy when compared to the GT in the three sections under the study.

### 5.6 Quantitative results

Figures 5 and 6 presents the visualization of the DA neuron segmentation results in test images using our best model. As seen, our method can effectively detect and segment the dopaminergic neurons of varying size and shape. Our quantitative results in Table 1, together with the qualitative results in Figures 5

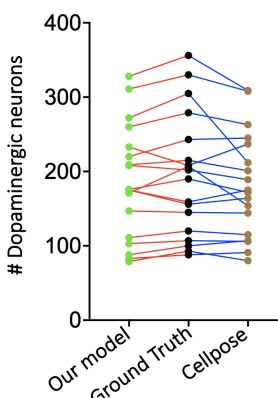

| ANOVA Summary | |
|---|---|
| F | 0.1309 |
| P-value | 0.8776 |
| P-value summary | n.s. |
| Significant diff. in means (p<0.05) | No |
| $R^2$ | 0.005108 |

| Bonferroni's multiple comparisons test | Mean Diff. | 95% CI of diff. | Below threshold | Summary | Adjusted P-value |
|---|---|---|---|---|---|
| Our model vs. GT | -8.556 | -67.54 to 50.43 | No | n.s. | >0.9999 |
| Our model vs. Cellpose | 4.278 | -54.71 to 63.26 | No | n.s. | >0.9999 |

Figure 4: Comparison of DA neuron counting between our model and the Cellpose baseline in individual sections. The green, black, and brown dots depict the cells counted by our model, ground truth (GT), and Cellpose, respectively. The red lines indicate the comparison between GT and our model. The blue lines indicate the comparison between GT and the Cellpose. Sections were selected from the blind dataset.

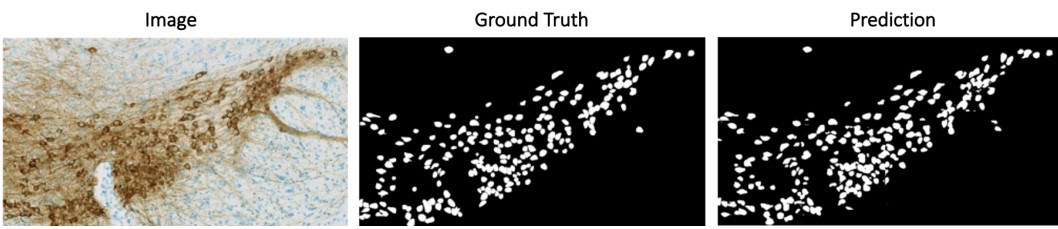

Figure 5: Visualization of Mouse brain 2D Image depicting DA neurons in the SN and segmentation results produced by our method.

and 6 demonstrate the capability of our framework in providing an effective solution for segmentation of dopaminergic neurons.

## 5.7   Analyzing phenotypic characteristics of DA neurons

**Experimental setup.** Extracting additional information on the state of the individual dopaminergic neurons, such as the TH intensity, can help biologists with additional data for analytic tasks in practical settings. In particular, the TH intensity is indicator of dopaminergic neuronal health. As such, measuring the TH intensity in individual cells can help us to categorize the cells in compartments defining the effect of disease pathology progression in different mouse models. The information about the health of the neurons is extremely valuable to determine the efficacy of drugs in PD preclinical trials. Studies have shown that TH intensity is lost in specific DA neurons in PD transgenic and injection animal models but the overall analysis of such observation manually is next to impossible (Kawahata & Fukunaga, 2020; Pajarillo et al., 2020; Vecchio et al., 2021). Thanks to our automated method for segmenting the DA neurons, we provide a measurement of individual cell TH intensities, facilitating deeper analysis into the biology of DA neuronal loss and understand the mechanism through which this loss happens and what it suggests in terms of PD pathogenesis. To do so, we measured the TH intensity after converting the images into grayscale (8-bit, 0-255 range). The lower the number or closer to 0, the darker the stain is. The higher the number or closer to 255, the lighter the stain. The TH intensity was measured on ImageJ, which is a platform for analyzing digital images.

**Results.** Figure 7 shows the TH intensity (brown color) of individual DA neuronal cell bodies in 5 different gradients. The gradient was obtained by measuring the TH intensity for an entire dataset and splitting it into 5 different groups and visual and numerical data was obtained for each neuron.

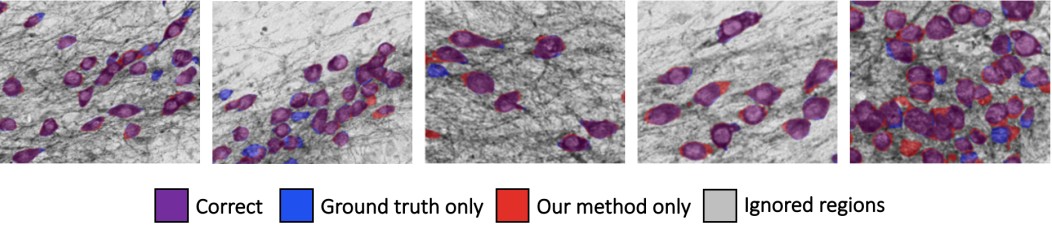

Correct    Ground truth only    Our method only    Ignored regions

Figure 6: Visualization of cell segmentation results yielded by our model compared with ground truth.

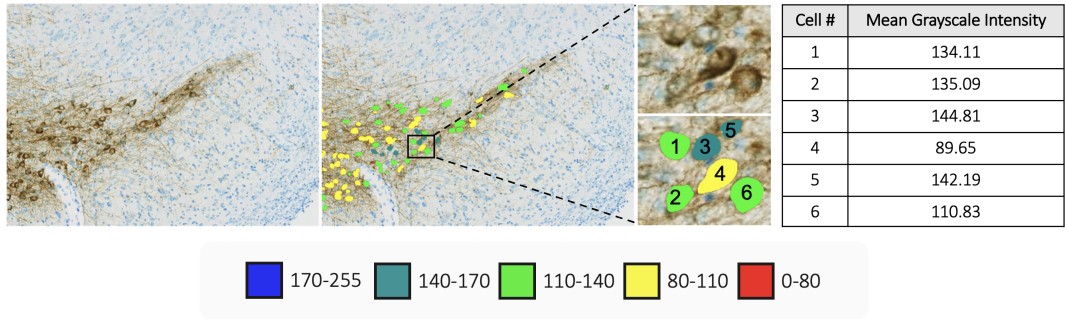

170-255    140-170    110-140    80-110    0-80

Figure 7: Measurement of cell TH intensity. The top panel shows an example image and the color overlay of mean intensity measured in 8-bit grayscale. The bottom panel shows the intensity color legend, a magnified view of several cells outlined in the black box above and a table of mean grayscale intensity value for each cell.

## 6 Ablation Experiments

**Experimental setup.** We conduct extensive ablation experiments on different data augmentation techniques and network architectures. We examine seven different combinations of transformation that are commonly used in the literature, including (1) no augmentation (mode 1), (2) Flip, Rotation, RandomBrightnessContrast, and RandomGamma (mode 2), (3) Flip, Rotation, RGBShift, Blur, GaussianNoise (mode 3), (4) Flip, Rotation, RGBShift, Blur, GaussianNoise, RandomResizedCrop (mode 4), (5) Flip, Rotation, RGBShift, Blur, GaussianNoise, RandomResizedCrop, Elastic Transformation (mode 5), (6) Flip, Rotation, RandomBrightnessContrast, RandomGamma, RGBShift, Blur, GaussianNoise, RandomResizedCrop (mode 6), and (7) Flip, Rotation, RandomBrightnessContrast, RandomGamma, RGBShift, Blur, GaussianNoise, RandomResizedCrop, Elastic Transformation (mode 7). For network architectures, we examine U-Net and DeepLabV3+. In ablation experiments, all models are initialized with SwAV pre-trained model and fine-tuned with 25% of data.

**Results.** Table 4 shows the results of different data augmentation techniques. According to these results, the lowest performance comes from mode 1 (no augmentation), highlighting that combining pre-training with data augmentation techniques yields more accurate segmentation results for downstream tasks with limited amounts of data. Additionally, the combination of Flip, Rotation, RGBShift, Blur, GaussianNoise, RandomResizedCrop (mode 4) provides the best performance among all data augmentation approaches. This implies that color transformations such as RGBShift, Blur, and GaussianNoise can help the deep model in gleaning more generalizable representations. Furthermore, a comparison of the results obtained by modes 3 and 4, the latter of which includes an additional RandomResizedCrop, reveals that random cropping significantly contributes to performance improvements. Moreover, a comparison of the results obtained by modes 4 and 5, the latter of which includes an additional elastic transformation, demonstrates that elastic transformation has a negative impact on performance; the same observation can be drawn from the comparison of modes 6 and 7.

| Mode | Data Augmentations | | | | | | | | | Dice(%) |
|---|---|---|---|---|---|---|---|---|---|---|
| | Flip | Rotation | Brightness Contrast | Gamma | RGB Shift | Blur | Gaussian Noise | Random Crop | Elastic | |
| 1 | - | - | - | - | - | - | - | - | - | 78.96±1.85 |
| 2 | ✓ | ✓ | ✓ | ✓ | - | - | - | - | - | 80.83±1.17 |
| 3 | ✓ | ✓ | - | - | ✓ | ✓ | ✓ | - | - | 80.64±1.06 |
| 4 | ✓ | ✓ | - | - | ✓ | ✓ | ✓ | ✓ | - | **81.94±0.74** |
| 5 | ✓ | ✓ | - | - | ✓ | ✓ | ✓ | ✓ | ✓ | 79.97±2.73 |
| 6 | ✓ | ✓ | ✓ | ✓ | ✓ | ✓ | ✓ | ✓ | - | 81.30±0.93 |
| 7 | ✓ | ✓ | ✓ | ✓ | ✓ | ✓ | ✓ | ✓ | ✓ | 80.43±1.18 |

Table 4: Comparison of different data augmentations.

| Network Architecture | Dice(%) |
|---|---|
| DeepLabV3+ | 81.53±0.76 |
| U-Net | **81.94±0.74** |

Table 5: Comparison of different segmentation architectures.

Table 5 presents the results of different network architectures for neuron segmentation task. As seen, U-Net, originally was designed for medical segmentation tasks, provides superior performance over DeepLabV3+.

# 7 Conclusion

In this paper, we developed a robust machine learning model that can detect and count the DA neurons reliably in independent animal studies. This is an immediate requirement in the field of PD research to accelerate the in-vivo screening of potential drugs so that more drugs can be taken into the clinic for human trials. The existing manual counting or stereology based method is unable to keep up with the number of studies currently conducted in different labs focusing on this area. Additionally, it also suffers from human bias which makes the data interpretation extremely cumbersome. The study framework is established on a self-supervised learning paradigm to combat the lack of large-scale annotated data for training deep models. We also realized that using segmentation based methods facilitated us to go beyond counting the number of DA neurons which the existing machine learning models are implementing. In addition to counting the DA neurons, we were able to obtain characteristics of the DA neurons which is very valuable to understand the health status of individual DA neurons on a histology slide. The loss of dopaminergic neurons determines the extent of pathogenesis in PD. Our model determines the number of dopaminergic neurons by counting the number of cell body/ soma of dopaminergic neurons. In addition, the ability to measure the TH intensity in individual cells (indicator of dopaminergic neuronal health) helps us to categorize the cells in compartments defining the effect of disease pathology progression in different mouse models. The information we obtain on the health of the neurons is extremely valuable to determine the efficacy of drugs in PD preclinical trials. Studies have shown that TH intensity is lost in specific DA neurons in PD transgenic and injection animal models but the overall analysis of such observation manually is next to impossible. With our method, we can get the cumulative data to dig deeper into the biology of DA neuronal loss and understand the mechanism through which this loss happens and what it suggests in terms of PD pathogenesis. There are additional challenges to consider such as limited datasets, tissue section thickness, image resolution and overlapping cells but our model has demonstrated very high efficiency taking into consideration all these factors. With the advancement in machine learning and biology, these models will improve and provide solutions to the ever increasing demand for data-analysis in research biology. Our experimental results suggests that our method can be extrapolated to other species that are used as animal models in PD. With the addition of more labeled data, we could go deeper in understanding the biology of DA neuronal loss by capturing the changes which are visible or sometimes not visible to the human eye. To summarize, this method will be very useful to shorten the time needed to analyze loss of DA neurons in animal studies and accelerate the drug discovery of PD.

## 8 Acknowledgements

We would like to thank Oded Foreman, Amy Easton and William J. Meilandt for providing the resources to facilitate this study. We appreciate the assistance from Kimberly Stark for animal study management and Renee Raman for imaging management. We would also like to thank Neuroscience Associates Inc. (Knoxville, TN, USA) for providing the tissue processing infrastructure to conduct this study. We express our gratitude to Labelbox Inc. for providing us the infrastructure to manage and label images. Labelbox is a prominent data-centric AI platform that empowers teams working with generative AI and large language models (LLMs) to infuse these systems with the optimal balance of human guidance and automation.

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
