# OpenReview forum: "Self-supervised Learning for Segmentation and Quantification of Dopamine Neurons in Parkinson’s Disease"
_TMLR — Accepted by TMLR_

### Review · Reviewer_coWh · 2023-05-25

**Summary Of Contributions:**

This paper presents an approach for segmenting and counting dopaminergic (DA) neurons in whole-slide digital pathology images.  The method is based on self-supervised pre-training first on ImageNet, followed by fine-tuning on microscopy images, and finally supervised learning. Experiments evaluated the effect on the fine-tuning stages as well as the performance on cell counting.

**Audience:**

Yes

**Broader Impact Concerns:**

There is no concerns on ethical implications of the work.

**Claims And Evidence:**

Yes

**Requested Changes:**


- The authors should consider clearly outlining the challenges that exist in existing cell segmentation and counting methods that preventing them from being directly applied to the DA neuron segmentation problem.

- The authors should consider including representatives about the above methods to demonstrate why the presented method is needed or favorable.

- The authors should add details on human annotators and annotating uncertainty.

- The authors should add statistical test for difference on the presented quantitative results.

- The authors should provide clarifications on the last two questions listed in the section above.


**Strengths And Weaknesses:**

Strengths:

- This paper seems to represent the first application of cell segmentation and counting DNNs for DA neurons.
- The experimentation provided some relatively comprehensive results on the effect of pre-training on generic versus in-domain dataset on downstream tasks


Weaknesses

- A major concern is the unclear methodological contribution of the presented work with respect to existing cell segmentation and counting approaches. It was indicated that these existing methods have limitations when applied on DA neurons in animal models of PD, but it was not clearly explained what may be the limitations and challenges, and how the presented works overcome these challenges. In another word, it is not clear why the presented approach can achieve what existing methodologies cannot achieve.

- A related concern to the above is the lack of considerations on alternative methods that can be applied to this problem during experimentation. Cell-counting was compared to Cellpose, but it was not clear what is the underlying methodology for Cellpose. More importantly, among existing cell segmentation and counting models, none of them seem to be considered. Again this is related to the comment above — the authors need to either explain why none of these methods can be applied to this problem, or include some of them as baseline methods in order to demonstrate the contribution of the presented method.

- Another major concern is the lack of discussion and consideration on the human annotator(s) and the potential labeling uncertainty associated with the human annotator(s). It was not clear how many biologists are involved as the human annotator(s). If multiple are involved, it should be discussed how inter-operator disagreements are addressed to reach consensus in labeling. If only one human annotator is involved, it should be explicitly described and limitation discussed.

- Many of the results presented, such as those in Table 1, seem to show differences that are not statistically significant (the gain in mean seems to be smaller than 1 std). This should be discussed, and statistical tests should be done for significance.

- For results on pre-training (Table-2), is there a reason that the effect of in-domain pre-training and imageNet—>In-domain pre-training is not tested on SwAW, considering that SwAW seems to give more favorable results on ImageNet alone?

- For the baseline for cell-counting, it was not clear what is the “naive approach of counting cells by the number of connected components”. Is it based on the segmentation results of the presented approach, but only differ in not using the method described in section 3.7? Or is it an entirely different approach?

---

> ### Author Response · Authors · 2023-07-01
> **Responses**
>
> We appreciate your time and careful comments, and strive to respond to each and every comment as follows:
>
> __A major concern is the unclear methodological contribution of the presented work with respect to existing cell segmentation and counting approaches. It was indicated that these existing methods have limitations when applied on DA neurons in animal models of PD, but it was not clearly explained what may be the limitations and challenges, and how the presented works overcome these challenges. In another word, it is not clear why the presented approach can achieve what existing methodologies cannot achieve.__
>
> The existing methods to count dopaminergic neurons in histology images rely on nucleus detection and optical density of the staining used to detect them. DA neurons are specialized neurons and tend to be very dense in substantia Nigra (SN).  SN is the anatomical region where PD pathology leads to loss of DA neurons. Due to the high density of these neurons, the OD based nucleus detection fails to detect the number of neurons accurately. Additionally, based on the nucleus detection method, we cannot analyze the loss of TH staining in the cell body/ soma which indicates DA neuronal health. Our methodology can segment the soma/ cell body of individual DA neurons which provides us the number of DA neurons and status of its health. The information we obtain on the health of the neurons is extremely valuable to determine the efficacy of drugs against PD in preclinical trials and will enable us to learn about the biology of dopaminergic neurons in PD pathogenesis. Literature cites that TH intensity is lost in specific DA neurons in PD transgenic and injection animal models but the overall analysis of such observation manually is next to impossible. With our method, we can get the cumulative data to dig deeper into the biology of DA neuronal loss and understand the mechanism through which this loss happens and what it suggests in terms of PD pathogenesis. This information will lead to discovery of new drug targets and better preclinical trial outcomes.
>
>
> __A related concern to the above is the lack of considerations on alternative methods that can be applied to this problem during experimentation. Cell-counting was compared to Cellpose, but it was not clear what is the underlying methodology for Cellpose. More importantly, among existing cell segmentation and counting models, none of them seem to be considered. Again this is related to the comment above — the authors need to either explain why none of these methods can be applied to this problem, or include some of them as baseline methods in order to demonstrate the contribution of the presented method.__
>
> Besides comparing our method with regular U-Net models for segmentation, which rely on either random initialization or supervised ImageNet pretrained models, we have compared our method with Cellpose, which is the most advanced platform available to segment cells. It is a generalized cell segmentation model and our model specializes in DA neurons.
>
> Cellpose is a state-of-the-art generalist model that can segment many types of cell without requiring new training data or further model retraining. Cellpose relies on a large dataset of varied images of cells and a reversible transformation from training set masks to vector gradients that can be predicted by a neural network. Cellpose leverages a U-Net model to predict the horizontal and vertical gradients, as well as whether a pixel belongs to any cell. The three predicted maps are combined into a gradient vector field. In the test time, the predicted gradient vector fields are used to construct a dynamical system with fixed points whose basins of attraction represent the predicted masks.
>
> We have now added the details of the Cellpose method in the related works section.
>
> __Another major concern is the lack of discussion and consideration on the human annotator(s) and the potential labeling uncertainty associated with the human annotator(s). It was not clear how many biologists are involved as the human annotator(s). If multiple are involved, it should be discussed how inter-operator disagreements are addressed to reach consensus in labeling. If only one human annotator is involved, it should be explicitly described and limitation discussed.__
>
> We have evaluated human annotator bias in one of our previous neuroanatomy segmentation studies that have been published. We observed a 5-8% difference in output with different manual annotators. Since manual annotation and counting of individual cells is extremely laborious in a data-set used in this study (our data-set had approximately 30,000 individual cells), to mitigate the human annotator bias, we had 3 annotators who randomly annotated the cells for the GT in this study. A neuroanatomy expert further QCd the annotations and made the necessary changes to reflect the cell annotation. The representative image is depicted in Fig 2 ground truth panel.

---

> ### Author Response · Authors · 2023-07-01
> **Responses - continued**
>
> __Many of the results presented, such as those in Table 1, seem to show differences that are not statistically significant (the gain in mean seems to be smaller than 1 std). This should be discussed, and statistical tests should be done for significance.__
>
> We have now provided the statistical analyses based on an independent two-sample t-test for the results in Table 1 and Table 2, and included the results in the paper.
> * In Table 1, our statistical analysis demonstrates that when using  full training data, the best self-supervised method (i.e SwAV) yields significant boost (p-value= 0.00171) compared with the best baseline (i.e. training from random initialization). Furthermore, when using partial training data, the best self-supervised method (i.e SwAV) yields significant boost (p-value=0.00738) compared with the best baseline (i.e. supervised ImageNet model).
> * In Table 2, our statistical analysis reveals that pretraining on ImageNet followed by pretraining on In-domain data provides significant boosts compared with both in-domain pretraining (p-value= 0.00006) and ImageNet pretraining standalone (p-value=0.00398).
>
> __For results on pre-training (Table-2), is there a reason that the effect of in-domain pre-training and imageNet—>In-domain pre-training is not tested on SwAW, considering that SwAW seems to give more favorable results on ImageNet alone?__
>
> We chose Barlow Twins as a proof of concept to demonstrate the efficacy of two-stage pretraining for dopamine neuron segmentation, primarily due to its simplicity and computational efficiency.
> * Barlow Twins is renowned for its simplicity compared to other self-supervised methods, utilizing a straightforward framework that relies on a cross-correlation loss for learning representations. This simplicity makes it relatively easier to train and optimize.
>
> * Additionally, Barlow Twins is designed to be computationally efficient by avoiding computationally expensive operations like contrastive or clustering-based methods. As such, it becomes advantageous in scenarios with limited computational resources.
>
>
> __For the baseline for cell-counting, it was not clear what is the “naive approach of counting cells by the number of connected components”. Is it based on the segmentation results of the presented approach, but only differ in not using the method described in section 3.7? Or is it an entirely different approach?__
>
> A naïve method of counting cells from segmentation predictions involves computing the connected components within the masks and considering the number of connected components as the cell count. However, this approach may not be accurate due to the presence of overlapping cells that share boundaries. To address this issue and improve counting accuracy, we propose a post-processing technique outlined in Section 3.7. The effectiveness of our approach in enhancing cell counting accuracy compared to the naive approach is demonstrated in the ablation study presented in Table 3.b.
>
> We have now revised Sec. 3.7 to further elaborate this.
>
> __The authors should consider clearly outlining the challenges that exist in existing cell segmentation and counting methods that preventing them from being directly applied to the DA neuron segmentation problem.__
>
> Certainly. In the revised version, we have provided a comprehensive introduction that clearly outlines the challenges associated with applying existing cell segmentation models to the task of dopaminergic neuron segmentation. This explanation aims to emphasize why a direct application of these models is not feasible.
>
> __The authors should consider including representatives about the above methods to demonstrate why the presented method is needed or favorable.__
>
> Studies in literature who have attempted to perform this task depict how the detection and counting of dopaminergic cells is done when nucleus and optical density is used to detect the cells. (Fig 1, PMID: 30144349). This is the most advanced platform available to detect DA neurons. Cellpose is an advanced platform and we have compared our model to that one here. The representative figures in Fig 4 and 7 of our manuscript show how our model with the highest Dice %  is identifying the DA neurons and thereby counting them.  Regarding the different models represented in Table 4, we relied on Dice% as the output measure for how the model performs. We chose to perform the task with the model showing the highest Dice %. The goal of the study was to compare our best model to other available models trying to perform the same task. Along those lines, we unfortunately did-not collect the images from our less efficient models (based on Dice %). With additional time, we could attempt to obtain those images but we believe it would-not provide additional value to this study.

---

### Review · Reviewer_8wRy · 2023-06-14

**Summary Of Contributions:**

This submission presented a system that automatically count the number of dopaminergic neurons. The system consist of a segmentation model that finetune a pretrained model on digital microscopy images with manual segmentation annotations. The paper studied the source of the pretrained model in terms of pretraining methods (supervised pretraining and a few self-supervised algorithms), pretraining data domain. The paper found that in the 25% fine-tuning data setting, self-supervised pretraining on ImageNet and then in-domain data make the best segmentation model. More ablation studies on data augmentation choices and network architectures are provided. The counting results from the deep learning system is compared with the manual counts as well as a generalist cell segmentation model, Cellpose.

**Audience:**

No

**Broader Impact Concerns:**

There is not a Broader Impact statement and I don't think it need one.

**Claims And Evidence:**

No

**Requested Changes:**

Besides the weakness mentioned above, there are few more suggestions for improving the paper.
- For the ablation on data augmentations, it is easier to have table of checking marks for each augmentations for more clear comparison of each mode
- It would be more convincing to show how the  segmentation performance changes translate into counting errors changes.

**Strengths And Weaknesses:**

Strengths:

- The paper is written well and organized.
- The paper studied many design choices and provided a good view on how each component contribute to the final results.

Weakness:

- The machine learning related contribution in the paper is the segmentation system on the specific domain of DA neurons in histopathology images. The observations the presented for transfer learning e.g. pretrained model is better than random initialization, additional pretraining on in domain data is better than the image-net pretrained one, et al. have been seen in other studies of different computer vision applications. Although I believe this paper is valuable and interesting for researchers in the medical imaging and histopathology domains, I am not sure how interesting it will be for TMLR audience.
- Some clarification and details are needed for better evaluated the correctness of the paper
    - Authors claims that the system “provides the phenotypic characteristics of individual dopaminergic neurons as a numerical output”. What exactly are phenotypic characteristics? Does it refer to the mean intensities of cells or anything else?
    - In the qualitative results, the paper states that“In Figure 6, the statistics shows there is not significant changes between the DA neurons counted by the model or Cellpose when compared to GT. Deeper analysis into the data shows that Cellpose had a significant difference from GT in three sections but our model was able to detect the DA neurons with higher accuracy.” How should we understand this? They sounds contradictory? Which “three sections” refer to?
    - There are two different descriptions for how to build the test set:  1.“For supervised learning, we randomly divided the dataset into training (70%), validation (10%), and testing (20%).” 2. “The blind test data-set used for analyzing model’s efficiency was a separate animal study in which the model has not been directly trained on the study group.” Which one is the correct one, random split or a separate study?

---

> ### Author Response · Authors · 2023-07-01
> **Responses**
>
> We appreciate your time and careful comments, and strive to respond to each and every comment as follows:
>
> __The machine learning related contribution in the paper is the segmentation system on the specific domain of DA neurons in histopathology images. The observations the presented for transfer learning e.g. pretrained model is better than random initialization, additional pretraining on in domain data is better than the image-net pretrained one, et al. have been seen in other studies of different computer vision applications. Although I believe this paper is valuable and interesting for researchers in the medical imaging and histopathology domains, I am not sure how interesting it will be for TMLR audience.__
>
> Thanks for your comment. We believe that this study would be interesting for TMLR audience because:
>
> * This is the first study where the soma of dopaminergic neurons are identified and segmented by machine learning platform
> * We are able to measure TH intensity in the cell body of individual DA neurons (indicator of its health) and compartmentalize them based on that to answer deeper biology questions in PD.
> * Generally, tasks of this type lack a large data-set and we were able to apply SSL to obtain the cell counts with a high accuracy.
>
> We believe the novelty of this study lies in the application of machine learning pipeline to efficiently execute specialized biological tasks which otherwise is time consuming, prone to human labeler bias and not accurate when compared to human labeller.
>
> __Authors claims that the system “provides the phenotypic characteristics of individual dopaminergic neurons as a numerical output”. What exactly are phenotypic characteristics? Does it refer to the mean intensities of cells or anything else?__
>
> The loss of dopaminergic neurons determines the extent of pathogenesis in PD. Our model determines the number of dopaminergic neurons by counting the no. of cell body/ soma of dopaminergic neurons. It is one of the most important phenotypic characteristics in PD. In addition, the ability to measure the TH intensity in individual cells (indicator of dopaminergic neuronal health) helps us to categorize the cells in compartments defining the effect of disease pathology progression in different mouse models. The information we obtain on the health of the neurons is extremely valuable to determine the efficacy of drugs in PD preclinical trials. Studies have shown that TH intensity is lost in specific DA neurons in PD transgenic and injection animal models but the overall analysis of such observation manually is next to impossible. With our method, we can get the cumulative data to dig deeper into the biology of DA neuronal loss and understand the mechanism through which this loss happens and what it suggests in terms of PD pathogenesis. This information will lead to discovery of new drug targets and better preclinical trial outcomes. We have put an effort to measure the shape and area of individual cells as phenotypic characteristics but the intrinsic variability of shape and size of dopaminergic neurons in 35 micron brain sections led to variable output. Hence, these two features are out of scope of this study.
>
> __In the qualitative results, the paper states that“In Figure 6, the statistics shows there is not significant changes between the DA neurons counted by the model or Cellpose when compared to GT. Deeper analysis into the data shows that Cellpose had a significant difference from GT in three sections but our model was able to detect the DA neurons with higher accuracy.” How should we understand this? They sounds contradictory? Which “three sections” refer to?__
>
> Sorry for the confusion. We have corrected the sentence and the Figure. Each black dot in the image is a section, It is observed that the three arrows indicate three sections where Cellpose determines the number of DA neurons lesser than the GT. Our model does a better job in measuring the number of neurons when compared to Cellpose. The sections are from blind data-set as explained in section 1.3.

---

> ### Author Response · Authors · 2023-07-01
> **Responses - continued**
>
> __There are two different descriptions for how to build the test set: 1.“For supervised learning, we randomly divided the dataset into training (70%), validation (10%), and testing (20%).” 2. “The blind test data-set used for analyzing model’s efficiency was a separate animal study in which the model has not been directly trained on the study group.” Which one is the correct one, random split or a separate study?__
>
> We revised section 1.3 to clarify our splits and test sets. The first test set, which we call the in-study test set, is 20% of the data that was not used for either self-supervised learning (SSL) or fine-tuning. The second test set, which we call the out-study test set, consists of 20 brain sections randomly selected from a different animal study with 150 brain sections. The ground truth (GT) for the out-study test set was established by annotating the soma. Our model and Cellpose were tested on these 20 sections, and the results are shown in Figures 4-6. The model had never seen these sections before.
>
> __For the ablation on data augmentations, it is easier to have table of checking marks for each augmentations for more clear comparison of each mode__
>
> Thank you for your suggestion; we have revised the table to make it more readable.

---

### Review · Reviewer_Chm3 · 2023-06-16

**Summary Of Contributions:**

The paper explores different self-supervised learning strategies for segmenting and quantifying Dopamine Neurons on histopathology images. The strategies explored in the paper include SwaV, Barlow Twins, Deep Cluster, and supervised pretrain on ImageNet. The paper demonstrate that the best-performing strategy steps are 1) SSL pretraining on ImageNet using BarlowTwins, 2) SSL training on the medical data using Barlow Twins, 3) fine tuning with the limited labeled dataset.

**Audience:**

No

**Broader Impact Concerns:**

I don't see any ethical implications of the work.

**Claims And Evidence:**

Yes

**Requested Changes:**

The paper applies existing SSL methods for segmenting and quantifying Dopamine Neurons on histopathology images and does not present a methodological contribution. Therefore, I believe it is more suitable for an applied venue. Please see my comments about the design of the experiments above for potential submission to such a venue.

**Strengths And Weaknesses:**

Strengths:
- The analysis of different SSL strategies is interesting. The paper provides a nice insight into exploiting SSL strategies for a particular task in a limited labeled setting.

Weaknesses:
- Novelty: I think novelty is the main weakness of the paper. The paper evaluates the combinations of different SSL strategies a medical imaging task; however, it doesn't propose a technical contribution.
- Shortcoming of the experimental design: I think there are a couple of crucial shortcomings of the paper due to experimental design.
1- The experimental results were presented on only a single in-house dataset. Therefore, it is difficult to generalize the paper's conclusions to different datasets, even for the same problem. I would expect to see experiments on multiple datasets to reach the conclusions drawn in the paper. Additionally, using only an in-house dataset restricts the reproducibility of the experiments.
2- In Sec 3.3., the paper mentions that there are 1500 images in total, and "all" images are used in the SSL training. Training/test/val splits are only mentioned for the supervised fine-tuning part. As far as I understand, the test images used in the supervised training part were also used in the SSL part. This is quite unusual to me since the model should not see the test images at any stage of the training, even during the unsupervised one.
3- The paper mentions some methods for neuron counting in the related work section but doesn't present any comparison.

---

> ### Author Response · Authors · 2023-07-01
> **Responses**
>
> We appreciate your time and careful comments, and strive to respond to each and every comment as follows:
>
> __Novelty: I think novelty is the main weakness of the paper. The paper evaluates the combinations of different SSL strategies a medical imaging task; however, it doesn't propose a technical contribution.__
>
> Thank you very much for your feedback. We believe that the novelty of this study are the followings:
>
> * This is the first study where the soma of dopaminergic neurons are identified and segmented by machine learning platform
> * We are able to measure TH intensity in the cell body of individual DA neurons (indicator of its health) and compartmentalize them based on that to answer deeper biology questions in PD.
> * Generally, tasks of this type lack a large data-set and we were able to apply SSL to obtain the cell counts with a high accuracy.
>
> We believe the novelty of this study lies in the application of machine learning pipeline to efficiently execute specialized biological tasks which otherwise is time consuming, prone to human labeler bias and not accurate when compared to human labeller.
>
> __Experimental design (1)- The experimental results were presented on only a single in-house dataset. Therefore, it is difficult to generalize the paper's conclusions to different datasets, even for the same problem. I would expect to see experiments on multiple datasets to reach the conclusions drawn in the paper. Additionally, using only an in-house dataset restricts the reproducibility of the experiments.__
>
> Sorry for the confusion. Our dataset includes 1500 histology images from 10 different internal animal studies. These studies were done at different timepoints and collected together to create the data-set for this study. These are the only datasets we have, and we do not possess any others. The goal of our work is to identify the soma of TH cells in the substantia nigra. TH cells are a type of neuron that produces the neurotransmitter dopamine. There is no publicly available dataset that closely resembles the nature of our dataset.Therefore, we assure that there will be no concerns regarding the reproducibility of the experiments.
>
> __Experimental design (2)-  In Sec 3.3., the paper mentions that there are 1500 images in total, and "all" images are used in the SSL training. Training/test/val splits are only mentioned for the supervised fine-tuning part. As far as I understand, the test images used in the supervised training part were also used in the SSL part. This is quite unusual to me since the model should not see the test images at any stage of the training, even during the unsupervised one.__
>
> We removed section 3.3 which was confusing and instead revised section 1.3 to include all of the studies we have used for training, validation, and test. The first test set, which we call the in-study test set, is 20% of the data that was not used for either self-supervised learning (SSL) or fine-tuning. The second test set, which we call the out-study test set, consists of 20 brain sections randomly selected from a different animal study with 150 brain sections. The ground truth (GT) for the out-study test set was established by annotating the soma. Our model and Cellpose were tested on these 20 sections, and the results are shown in Figures 4-6. The model had never seen these sections before.
>
> __Experimental design (3)- The paper mentions some methods for neuron counting in the related work section but doesn't present any comparison.__
>
> Literature shows how the detection and counting of dopaminergic cells is done when nucleus and optical density is used to detect the cells. (Fig 1, PMID: 30144349). Cellpose is the most advanced platform available to detect specialized DA neurons. Cellpose is an advanced machine learning based platform and we have compared our model to that one in our paper. The representative figures in Fig 4 and 7 of our manuscript show how our model with the highest Dice %  is identifying the DA neurons and thereby counting them.
> Regarding the different models represented in Table 4, we relied on Dice% as the output measure for how the model performs. We chose to perform the task with the model showing the highest Dice %. The goal of the study was to compare our best model to other available models trying to perform the same task. Along those lines, we unfortunately did-not collect the images from our less efficient models (based on Dice %). With additional time, we could attempt to obtain those images but we believe it would-not provide additional value to this study.

---

> > ### Author Response · Authors · 2023-07-01
> > **Responses - continued**
> >
> > __The paper applies existing SSL methods for segmenting and quantifying Dopamine Neurons on histopathology images and does not present a methodological contribution.__
> >
> > Thank you very much for your feedback. We believe that this study would be interesting for TMLR audience because:
> >
> > * This is the first study where the soma of dopaminergic neurons are identified and segmented by machine learning platform
> > * We are able to measure TH intensity in the cell body of individual DA neurons (indicator of its health) and compartmentalize them based on that to answer deeper biology questions in PD.
> > * Generally, tasks of this type lack a large data-set and we were able to apply SSL to obtain the cell counts with a high accuracy.
> >
> > Based on the criteria/ highlighted on the TMLR journal and Action Editor accepting the manuscript to go through revision, we think this manuscript will be of interest to TMLR audience: ``Nor should it form the basis for rejecting work on a method considered not “novel enough”, as novelty of the studied method is not a necessary criteria for acceptance. We explicitly avoid these terms (“significant”, “impactful”, “novel”), and focus instead on the notion of “interest”. If the authors make it clear that there is something to be learned by some researchers in their area from their work, then the criteria of interest is considered satisfied.''

---

### Review · Reviewer_H8SG · 2023-06-24

**Summary Of Contributions:**

In this paper, the authors proposed a novel method based on self-supervised learning for segmentation and quantification of dopamine neurons in Parkinson’s disease. By leveraging info extracted from natural images and medical data, they use self-supervised pre-training to transfer these knowledge to a fine tuned supervised deep neural nets on labeled medical data. Comprehensive ablation studies were conducted to show contributions of each component of the proposed method.

**Audience:**

Yes

**Broader Impact Concerns:**

1. This paper is more like an application paper which apply self-supervised deep learning for segmentation and quantification of dopamine neurons. I do not see significant innovation regarding methodology. So it is more suitable for journals in the medical imaging area, e.g. IEEE transactions on medical imaging.

**Claims And Evidence:**

Yes

**Requested Changes:**

See comments on weaknesses.

**Strengths And Weaknesses:**

Strengths:
1. It looks novel and promising to leverage natural images and medical data to improve segmentation and quantification of medical images using self-supervised learning.
2. Ablation studies are pretty solid and comprehensive to reveal characteristics of the proposed method.

Weaknesses:

1. In the experiment section, my major concern is that it lacks comparisons with SOTA methods. Ideally, it is encouraged to have comparisons with SOTA methods from both traditional medical image segmentation area and deep learning based methods. So it will show improvement and advances of the proposed method in this area.
2. Because medical images show quite different characteristics	regarding imaging modalities, object shapes, resolution, colors, etc., I am wondering why using natural images for self-supervised pre-training helps the medical image segmentation problem. Any rationale behind it the authors could illustrate more?
3. In sec. 3.1, what is the number of histology digital images used in this study? I read 30k TH positive DA neurons.
4. Sec. 4, the authors observed self-supervised models have better performance than the supervised ImageNet model. Usually on the same dataset and prediction tasks, supervised method should have better performance than self-supervised learning methods. The author may want to clarify rationale behind their observations.
5. What are the major difference between self-supervised learning methods employed in this paper and the contrastive learning?
6. In page 8, the authors claim that “color transformations such as RGBShift, Blur, and GaussianNoise can help the deep model in gleaning more generalizable representations ”. I am wondering whether there are any shape deformations which can help to this problem? At least these neurons show oval shapes which may have some implications.

---

> ### Author Response · Authors · 2023-07-01
> **Responses**
>
> We appreciate your time and careful comments, and strive to respond to each and every comment as follows:
>
> __In the experiment section, my major concern is that it lacks comparisons with SOTA methods. Ideally, it is encouraged to have comparisons with SOTA methods from both traditional medical image segmentation area and deep learning based methods. So it will show improvement and advances of the proposed method in this area.__
>
> The existing methods to count dopaminergic neurons in histology images rely on nucleus detection and optical density (OD) of the staining used to detect them. DA neurons are specialized neurons and tend to be very dense in substantia Nigra (SN).  Due to the high density of these neurons, the OD based nucleus detection fails to detect the number of neurons accurately. Based on years of experience in DA neuronal counting,  we have seen that the centroid methods are very inaccurate. Stereology on the other hand is quite accurate but requires hours and is subjected to user associated bias.
>
>
> __Because medical images show quite different characteristics regarding imaging modalities, object shapes, resolution, colors, etc., I am wondering why using natural images for self-supervised pre-training helps the medical image segmentation problem. Any rationale behind it the authors could illustrate more?__
>
> We observed that transfer learning from self-supervised ImageNet models boosts the accuracy of our model significantly.  As highlighted in Sec. 4.1, we attribute this to the fact that self-supervised pre-trained models, in contrast to supervised pre-trained models, encode low/mid level features that are not biased to task-relevant semantics, thus generalizing better to the applications with significant domain shift compared with the pretraining dataset.
>
> __In sec. 3.1, what is the number of histology digital images used in this study? I read 30k TH positive DA neurons.__
>
> We have thoroughly revised section 1.3 to clarify the data we have used in our study. We integrated a dataset of 1500 digital microscopy images acquired from multiple in-house studies of mouse models of Parkinson's disease (PD). 108 of these images were manually annotated, resulting in 30,000 segmented TH positive DA neurons. We used the unannotated images for self-supervised learning and then fine-tuned the self-supervised pre-trained models with the annotated images (supervised learning). 20% of the development set was set aside as an in-study test set, while the remaining data was randomly divided into training (85%) and validation (15%). Additionally, one separate animal study consisting of 150 sections was included in this dataset.
>
> __Sec. 4, the authors observed self-supervised models have better performance than the supervised ImageNet model. Usually on the same dataset and prediction tasks, supervised method should have better performance than self-supervised learning methods. The author may want to clarify rationale behind their observations.__
>
> Apologies for any confusion.
>
> Recently, there have been significant advancements in self-supervised learning, specifically in the context of instance discrimination tasks. These methods have made remarkable progress in closing the performance gap with supervised pre-training and have even surpassed supervised pre-training in certain vision tasks. The great success of self-supervised learning in vision tasks has motivated us to explore its generalizability to the dopamine neurons segmentation task.
>
> In line with previous studies, we have observed that self-supervised learning techniques exhibit greater generalizability compared to supervised ImageNet models, which have traditionally been a popular choice for transfer learning in the field of medical imaging. By leveraging self-supervised learning methods, we have been able to achieve promising results in our domain-specific task, further highlighting the potential of self-supervised approaches for advancing medical imaging tasks. Our two-stage pre training scheme further advances the self-supervised ImageNet modes by tailoring them to our data.

---

> ### Author Response · Authors · 2023-07-01
> **Responses - continued**
>
> __What are the major difference between self-supervised learning methods employed in this paper and the contrastive learning?__
>
> Barlow Twins minimizes redundancy in the representations by measuring the cross-correlation matrix between the embeddings of two identical networks and making it close to the identity matrix. SwAV simultaneously clusters the images while enforcing consistency between cluster assignments produced for differently augmented views of the same image. Deep Cluster v2 uses a clustering loss to learn representations that are more discriminative between different objects. These methods are different from contrastive learning, which considers the augmented views of the same image as positive pairs and the ones from the other images as negative pairs, and aim to enhance the similarity between positive pairs while decreasing the similarity between negative pairs. Barlow Twins, SwAV, and Deep Cluster v2 do not rely on many negative samples, and still can learn representations that are more informative and less redundant than contrastive learning.
>
> We have now revised the related work section to further explain the differences between different self-supervised approaches.
>
> __In page 8, the authors claim that “color transformations such as RGBShift, Blur, and GaussianNoise can help the deep model in gleaning more generalizable representations ”. I am wondering whether there are any shape deformations which can help to this problem? At least these neurons show oval shapes which may have some implications.__
>
> We attempted to use Elastic transformation, a form of shape deformation, for our task of segmenting dopaminergic neurons (see Table 4.a). However, we observed that it had a detrimental effect on the accuracy of the results. This could be attributed to the unique characteristics of dopaminergic neuron somas, which differ significantly from other cell types, such as nucleotides that have an oval shape.

---

> > ### Comment · Reviewer_H8SG · 2023-07-18
> > **Well noted with thanks!**
> >
> > N/A